# Reuniting and Endolymphatic Duct Macrophages: Localization and Possible Roles

**DOI:** 10.3390/audiolres15060160

**Published:** 2025-11-20

**Authors:** Elisa Vivado, Daniele Cossellu, Paola Perin

**Affiliations:** 1Department of Molecular Medicine, University of Pavia, 27100 Pavia, Italy; 2Department of Brain and Behaviour Science, University of Pavia, 27100 Pavia, Italy; 3Fondazione IRCCS Policlinico San Matteo, 27100 Pavia, Italy

**Keywords:** inner ear, macrophage, tissue clearing, otolith, hydrops, immunology of the ear, utriculo–endolymphatic valve, endolymphatic duct, reuniting duct, inflammation

## Abstract

**Background**: The inner ear hosts several macrophage populations. Endolymphatic sac macrophages can phagocytose otoconia, and spiral limbus macrophages express genes for fluid shear stress sensing and bone remodeling. Obstruction of endolymph flow by saccular otoconia could be linked to endolymphatic hydrops. Since macrophages are strongly affected by inflammatory status, a role for them in otolith removal could provide a link between inflammation and hydrops. However, the distribution of macrophages around the reuniting duct (RD) and endolymphatic duct (ED), which are narrow structures likely prone to blockage, remains unexplored. **Methods**: We performed tissue clearing and light-sheet imaging on rat temporal bones. Autofluorescence and immunolabeling for collagen IV, smooth muscle actin, and Iba1 were used to visualize inner ear structures, blood vessels, and macrophages. **Results**: The connective tissue layer underlying the RD extended from the cochlear spiral limbus. The RD and spiral limbus hosted a continuous microvascular network and macrophage population, comprising both ameboid and ramified cells; macrophages also surrounded the underlying vestibulocochlear artery (VCA). A separate macrophage population, continuous with that of the saccular connective tissue, was found around the endolymphatic sinus and utriculo–endolymphatic (Bast’s) valve; macrophage patterns changed in the vestibular aqueduct and endolymphatic sac. **Conclusions**: Macrophages are observed in positions consistent with potential roles in sensing luminal changes and in the clearance of obstructive material from the RD and ED; functional confirmation will require targeted experiments.

## 1. Introduction

Obstruction of endolymph flow by otoliths or otolith aggregates originating from the saccule has been proposed as one of the pathological mechanisms underlying endolymphatic hydrops and Ménière’s disease [1,2,3,4,5,6,7,8,9,10]. Blockage of the reuniting duct (also known as reunion duct, RD), which connects the saccule to the cochlear scala media [11], has been associated with cochlear hydrops [2], while obstruction [1,2] or backflow [8] at the endolymphatic duct (ED) has been associated with vestibular hydrops.

Most anatomical studies of the RD and ED have relied on high-resolution CT imaging (cone-beam or microCT) to infer soft tissue architecture from bone landmarks [4,12]; more recently, other volumetric imaging methods such as X-ray tomography [8] and tissue clearing [11] have enabled direct visualization of soft tissues in situ. Tissue clearing additionally permits cell type-specific labeling [13]. In the present study we employed a tissue clearing method that provides three-dimensional observation of the labyrinth [14] and associated immune structures [15,16,17] in order to visualize macrophage populations associated with RD and ED in a rat.

Besides hydrops, Ménière’s disease has also been associated with inflammation [18,19,20,21]. A single cell transcriptome study of the crista ampullaris identified macrophages expressing several genes associated with the disease [22], and a longitudinal study in Meniere’s patients has shown plasmatic cytokine profiles able to affect macrophage polarization [20]. Macrophages could therefore represent a link between inflammatory status and hydrops [22,23,24,25]. Animal studies have shown that macrophages are “siphoned” into the vestibular aqueduct upon osmotic imbalance [26], and stria vascularis macrophages contribute to endolymph production [27]. Macrophages located around the RD and ED could affect endolymph flow either by phagocytosing otoconia, as observed in the endolymphatic sac [28], or by altering duct patency and permeability when activated by inflammatory stimuli.

Here, we applied tissue clearing and light-sheet microscopy (as in [14]) to reconstruct both the RD and ED in the rat’s inner ear, together with associated vasculature and resident macrophages.

## 2. Materials and Methods

### 2.1. Animals

Experiments were performed on adult inbred Wistar rats of both sexes (Table 1). Five temporal bones from four rats were used. Data from three animals (R1, R3, R4) were previously used to build a volumetric atlas of the rat inner ear [14]. Sample R2 was part of the same batches but was not previously analyzed.

This study was carried out in accordance with the recommendations of Act 26/2014, Italian Ministry of Health. The protocol (number 155/2017-PR) was approved by the Italian Ministry of Health and University of Pavia Animal Welfare Office (OPBA). All efforts were made to minimize the number of animals used and animal suffering.

The image stacks used in this study were derived from the same samples previously employed for temporal bone marrow reconstruction [17] and inner ear atlas generation [14]; therefore, all sample treatments followed the protocols in [17]. Sample R2 was part of the same batches but was not previously analyzed. Tissue clearing was performed with a variant of iDISCO+ [29], which enables imaging of both the brain and temporal bone [15].

Fluorescent labeling was carried out as described in [17]. The primary antibodies used in the present paper were as follows: mouse anti-smooth muscle actin (SMA, Abcam (Cambridge, UK) amab7817, 1:200), rabbit anti-collagen IV (ColIV, Abcam ab6586, 1:200), and rabbit anti-Iba1 (WAKO 019–19,741, 1:200). Secondary antibodies (Invitrogen (Carlsbad, CA, USA), 1:200) included donkey anti-rabbit and anti-mouse conjugated with Alexa 488, 555, or 647. The ColIV antibody was not isoform-specific and labeled both vascular and nonvascular structures [30]; however, vessels could be easily distinguished. Cleared samples were imaged with a mesoSPIM light-sheet microscope [31] at the Wyss Center for Bio and Neuroengineering in Geneva, Switzerland, as in [17]. Voxel sizes for each sample are listed in Table 1.

### 2.2. Image Analysis

Segmentation of the inner ear labyrinth from image stacks was performed semiautomatically using FIJI [32] and ITK-SNAP [33] following the pipeline described in [14]. Three experts performed independent segmentations, and Fleiss kappa values [34] were calculated for macrophages and labyrinth structures as:$$\begin{array}{l} k=\frac{p_{0}-p_{e}}{1-p_{e}}\\ p_{0}=\frac{1}{N\cdot n\cdot (n-1)}\cdot (\sum_{i=1}^{N}\sum_{j=1}^{k}\;\;\;n_{ij}^{2}-N\cdot n)\\ p_{e}=\sum p_{j}^{2} \end{array}$$
where *N* was the number of voxels of the stack, *n* the number of observers, and *k* the number of categories in segmentation. The average kappa value was 0.92±0.04 for inner ear structures and 0.76±0.09 for macrophages. Segmentation and annotation of inner ear structures were based on autofluorescence signal at 488 and 647 nm, as described in [17]. In addition, the signals from SMA, ColIV, and Iba1 labeling were used to segment arteries, blood vessels, and macrophages, respectively. Segmented object volumes were calculated directly from ITK-SNAP voxel counts, scaled to voxel size. Distances were measured manually with the 3DSlicer Markups module, by placing markers along the measurement path.

### 2.3. Macrophage Populations

The volumetric segmentation pipeline was as follows:The RD, ED and their associated vessels were segmented manually with ITK-SNAP from autofluorescence or vascular labeling signal stacks.ROI datasets were extracted with a custom FIJI (1.54p) script.Datasets were applied as mask to Iba1 signal image stacks, and ROIs were dilated to include surrounding macrophages.Macrophages were segmented from masked image stacks in a semi-automated way by thresholding the Iba1 signal and refining manually.Macrophage count was performed using the ImageJ (Version 1.54p) 3D Object counter.

As a measure of macrophage morphology, solidity was calculated as a Vm/Vh ratio, where Vm was the macrophage volume (obtained as voxel count) and Vh the volume of its convex hull (i.e., of the smallest convex polyhedron that can contain the entire macrophage) [35]. This measure allowed us to distinguish ramified and ameboid cells (Figure 1).

The solidity calculation was performed as follows:Tif stacks containing macrophage segmentations were loaded into MATLAB (Version R2021b) and binarizedMacrophages were defined as objects using the *bwlabeln* function with 26 connectivityMacrophage volumes and convex hull volumes were measured with the *regionprops3* functionSmall objects (<100 voxels) were removed with the function *bwareaopen*Large outliers deriving from segmentation artifacts were removed with *rmoutlier*

The FIJI, MATLAB, and Python (Version 3.11)scripts are included as Appendix A. Inner ear segmentations and accompanying label descriptions are available upon reasonable request to the authors. 

## 3. Results

Macrophages were identified and segmented in three temporal bones, yielding a description of their distribution around inner ear ducts. Although the very small sample limits the general significance of the present results, this is the first description of macrophage distribution around the RD and ED. The subpopulation composition, roles, and interindividual variability of periductal macrophages will be addressed in further work.

### 3.1. RD

The rat RD (Figure 2) is a flattened ribbon connecting the cochlear duct to the saccule, which tapers to a narrow oval section measuring in our samples 95 ± 25 µm by 20 ± 3 µm (n = 4). Underneath the duct, a connective tissue layer is present (Figure 2(A1)), which emerges without any visible transition from the modiolar side of the spiral limbus and merges on the other side into the connective tissue lining the saccule (Figure 2(B1, B2); see also Figure 3 in [14]). In the three Iba1 samples, several macrophages were found to be embedded within the RD connective tissue, and a few were found to reach the perilymphatic space (Figure 2(A2,A3)).

The connective tissue below the RD is highly vascularized: The vestibulocochlear artery (VCA) is visible as a SMA+ tortuous vessel coursing through bone below the duct (Figure 2C) and branching into capillaries feeding the spiral limbus and RD (Figure 3), and saccule (Figure 4). The capillary network fed venules at the endolymphatic sinus which converged into the vein of the vestibular aqueduct (Figure 4).

In four of the five samples, the VCA branched into a straight vessel which left the connective tissue at the RD end (Figure 5), crossing the scala vestibuli and reaching the utricular limiting membrane, similar to an arachnoid trabecule. Macrophage populations were observed in association with both the tortuous and trabecular artery, and also with the RD (Figure 5).

### 3.2. ED

The ED proximal part forms the endolymphatic sinus, which connects to the saccule through a saccular duct and to the utricle through the utriculo–endolymphatic (Bast’s) valve, which is usually closed [36]. After Bast’s valve, the duct enters the vestibular aqueduct, and at the opposite end of the aqueduct it forms the endolymphatic sac within the dura. The rat ED and sac are similar to a human’s, albeit simpler [37], and the saccular duct only consists of a narrowing between saccule and sinus.

The ED is contacted in its sinus region by small veins draining the saccule, on the side opposite Bast’s valve (Figure 4), and upon entering the vestibular aqueduct it runs parallel to the vestibular aqueduct vein (VVA, Figure 6). In our Iba1 samples, the ED was surrounded by macrophages throughout its length (Figure 6). Three clusters could be distinguished: 1-around the sinus, ramified macrophages were seen around the perilymphatic side of the structure, including Bast’s valve; 2-within the aqueduct, a reticular network of macrophages surrounded the duct and vein in a single layer, forming a complete cuff; 3-an irregular layer was located in the aqueduct and perisaccular connective (this population was more difficult to delineate because it appeared more numerous in the endolymphatic sac, where background autofluorescence was stronger).

The macrophage cuffs surrounding the duct and vein were partially overlapping, and they may belong to the same population. In one sample where complete reconstruction of the vestibular aqueduct was possible, macrophages occupied 7% of its volume. Bone canalicules lined with macrophages connect the aqueduct to local bone marrow; in the present work, however, we focused on macrophages within the aqueduct, since the connections to bone marrow were addressed in a previous study [17].

### 3.3. Macrophage Features

In one sample we were able to identify and segment macrophages around most inner ear structures (Table 2). Volume and radius were similar among groups, except in macrophages associated with the VCA. Macrophages associated with the vestibular aqueduct, ED, and VVA displayed higher solidity. However, macrophage solidity varied considerably (from 0.12 to 0.96), even within populations, and differences in background and label intensity at 2 micron voxel size precluded an unbiased comparison of solidity among populations because thin branch visibility was affected by signal intensity. This issue affected volumes as well, since in crowded places cell separation could be unclear.

Macrophages were counted after segmentation of the Iba1 signal in a completely reconstructed sample.

## 4. Discussion

The inner ear immune system is emerging as a very complex system [24,38], and includes several macrophage populations with different origins and functions [22,23,24,25,26,27,28,39,40,41,42]. A recent single-cell transcriptomic analysis identified five macrophage subpopulations in the mouse cochlea [43], including CD206+ and CD206+CD163+ cells in the spiral limbus that express genes related to fluid shear stress and bone metabolism. In the present study, we describe Iba1+ cells associated with the RD and ED of a rat. Since Iba1 is a nonspecific macrophage marker [44], further experiments are needed to compare populations; however, the continuity between RD and spiral limbus connective tissue and microvascular network suggests that duct macrophages may belong to, or be closely related to, these previously described limbal populations.

Our anatomical reconstructions show that macrophages associated with the RD include a layer within the connective tissue lining the RD and perivascular macrophages located around the underlying VCA. Macrophage branches reached the perilymph in association to vessels or at the edge of the connective tissue ledge. These locations place macrophages at the interface between endolymph, perilymph, and vasculature, consistent with roles in monitoring the local microenvironment. It is very interesting to note that, even in our limited sample, we observed anatomical variability in the presence of a free vessel within the scala vestibuli, a feature reminiscent of structural heterogeneity described in the human inner ear [45]. The exposure to perilymph in the vestibular cisterna, in front of the oval window, and also their association to a freely mobile vessel could allow macrophages to sense both chemical and mechanical stimuli and link them to inflammatory responses (as has been observed in other systems [46,47]).

Besides sensing fluid flow and pressure, we can speculate that RD macrophages could act as phagocytes and free endolymph from debris, including otoconia. The latter are formed by a mineralized protein core [48] which is connected to a loose protein network [49]. Otoconia size is variable, but measures range from below 1 µm to up to 30 µm [50], and saccular otoconia appear smaller than utricular ones [51,52]. Therefore, a single otoconium appears unlikely to effectively block the RD, but a clump of otoconia could easily do it, and otoconial aggregates have been found in human inner ears in Meniére’s disease patients [7]. Dislodged otoconia spontaneously degrade in low-Ca2+ endolymph, and this is assumed to be the reason BPPV episodes spontaneously recover [53]. However, the dissolution time course for otoconia trapped in a small volume such as the RD is not known, and the degradation mechanisms for otolithic membrane fragments and for the proteinaceous network binding otoconia together are similarly not clear. In the endolymphatic sac, a population of macrophages has been found to access the sac lumen and act as phagocytes in regard to several objects, including otoconia [28]. In this line of reasoning, we may speculate that the alteration or disappearance of RD macrophages may lead to the accumulation of debris.

Macrophages were also observed along the ED and endolymphatic sinus, where they were more ramified than those at the RD. Their distribution formed distinct clusters around the sinus, the vestibular aqueduct, and the endolymphatic sac. In particular, macrophages formed a cuff surrounding both the duct and the accompanying vein within the aqueduct, suggesting a close anatomical association between venous drainage and immune surveillance in this region. Aqueductal macrophages are known to include a migrating component that enters the duct upon osmotic imbalances [26], and our findings extend these observations by providing a detailed baseline map of their spatial organization. In the dorsal root ganglion, macrophages display different features between arterial and venous compartments [54], and it is therefore possible that ED macrophages differ in function from RD macrophages. Moreover, if we assume otoconia are able to clog the ED, it would appear more likely for them to stop at the isthmus (which is the narrowest ED point) than at the valve, where they would simply not enter the ED. Since opening of the valve involves changes in the relative geometry of the utricle and endolymphatic sinus, which is contained in the perilymphatic space rather than in the aqueduct connective tissue, the role of macrophages around Bast’s valve appears puzzling, possibly involving local permeability or stiffness changes affecting sinus and/or utricle fluid flow.

The current work is limited to descriptive observations in a small number of rat samples, and functional roles can only be speculated but not inferred. Nonetheless, the strategic positioning of macrophages near the RD, ED, and associated vessels provides a foundation for future studies in larger cohorts aimed at testing whether these cells contribute to endolymph homeostasis, debris clearance, or pressure sensing.

## 5. Conclusions

The presence of macrophage populations around RD, ED, and their associated vessels suggests that they could affect endolymph circulation in the labyrinth. Macrophages associated with RD may sense local pressure variations and be involved in the removal of otoconia debris plugging the duct lumen. Macrophages associated with ED appear different and related to the venous rather than the arterial compartment. However, their role, especially around Bast’s valve, although intriguing, remains obscure.

## Figures and Tables

**Figure 1 audiolres-15-00160-f001:**
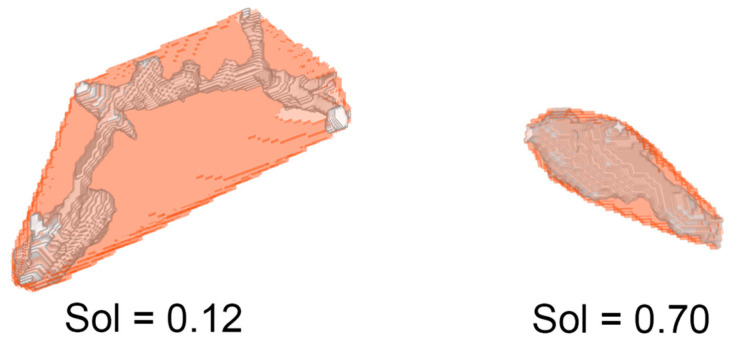
Example of macrophages with low solidity (0.12, **left**) and high solidity (0.70, **right**). Convex hull volume is in semitransparent orange, whereas the gray silhouette inside is the macrophage volume.

**Figure 2 audiolres-15-00160-f002:**
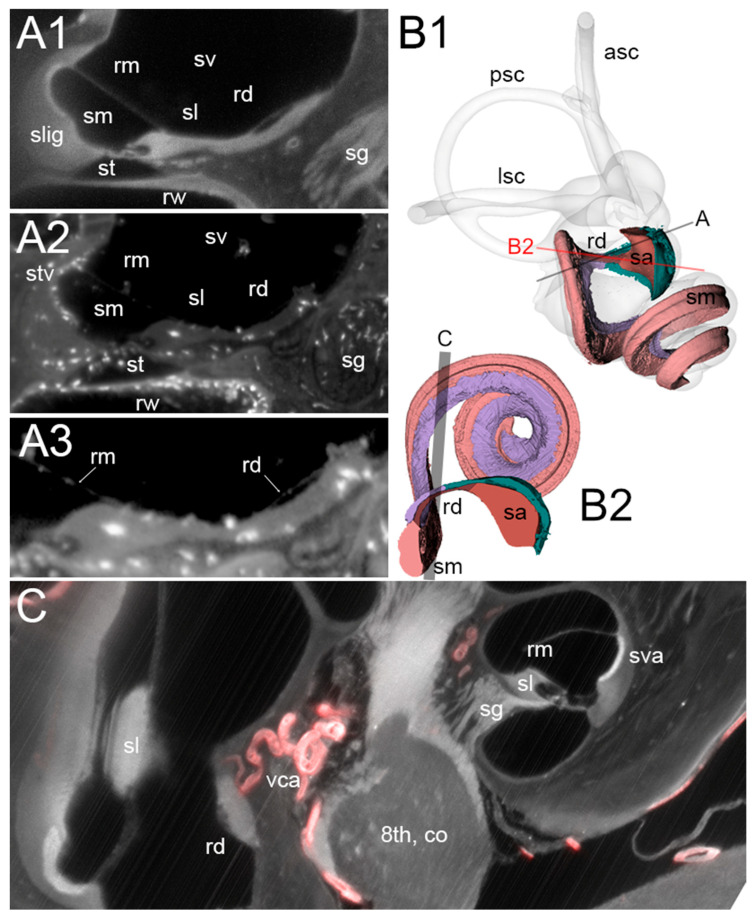
RD macrophages. (**A1**–**A3**): Single optical sections from autofluorescence signal (**A1**, sample R3) and Iba1 signal plus autofluorescence (**A2**,**A3**, sample R1b). The connective tissue under the RD is continuous with the spiral limbus and hosts macrophages that reach the perilymph of the scala vestibuli. (**A3**): magnified high-contrast detail of (**A2**) showing the thin membrane of RD and Reissner’s membrane. (**B1**,**B2**): Volumetric reconstruction of the RD in connection with the saccule (sa) and cochlear scala media (sm). (**B1**): labyrinth in its physiological orientation, as seen from a frontolateral point of view aligned with the anterior semicircular canal plane. Spiral limbus is shown in lilac, the connective tissue of the saccule in dark green. The overlayed lines display the orientation for panels A (black) and B2 (red). (**B2**): volumetric reconstruction cut to show RD longitudinal cross-section. The point of view is from the utricle to the cochlea. Shaded area indicates the stack in panel (**C**). (**C**): Maximal intensity projection of a 40 µm stack from autofluorescence (gray) and SMA signal (red), evidencing the location of VCA. Abbreviations: 8th, co: cochlear nerve; asc: anterior semicircular canal; lsc: lateral semicircular canal; psc: posterior semicircular canal; rd: reuniting duct; rm: Reissner’s membrane; rw: round window; sa: saccule; sg: spiral ganglion; sl: spiral limbus; slig: spiral ligament; sm: scala media; st: scala tympani; stv, sva: stria vascularis; sv: scala vestibuli, vca: vestibulocochlear artery.

**Figure 3 audiolres-15-00160-f003:**
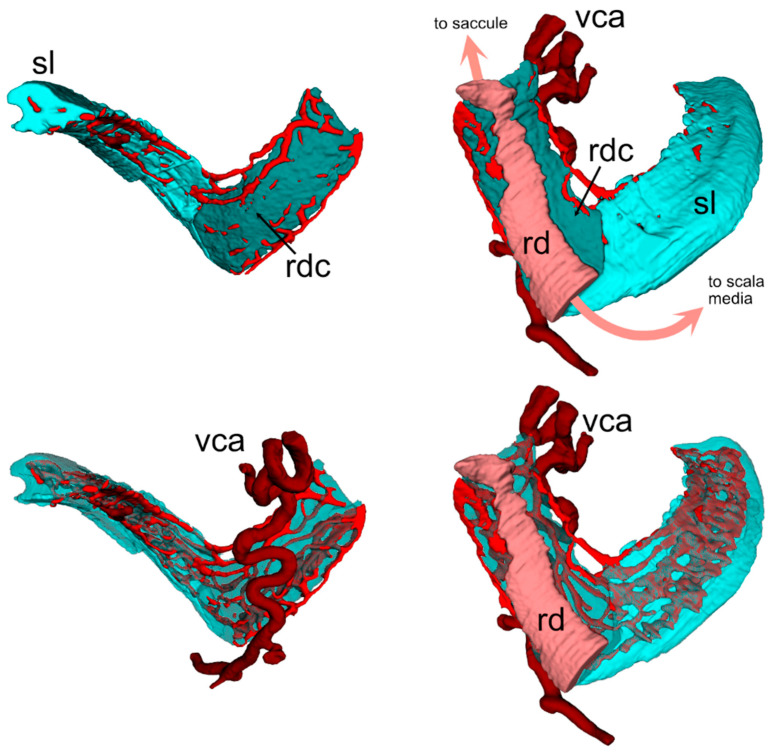
Volumetric reconstruction of the vascular network (from ColIV labeling) in the spiral limbus and RD. The connective tissue underlying the RD, shown in a darker color, displays no clear junction with the spiral limbus (the separation line was arbitrary, for annotation purposes). The capillary network branched from the VCA below the RD; higher within the cochlear duct spiral, the spiral artery in the modiolus fed into the same anastomosed network. Models on the right are turned 180 degrees versus models on the left. Models at the bottom are semitransparent to show the whole capillary network. Saccule and scala media are not shown for clarity. Abbreviations: rd: reuniting duct; rdc: RD connective tissue; sl: spiral limbus; vca: vestibulocochlear artery.

**Figure 4 audiolres-15-00160-f004:**
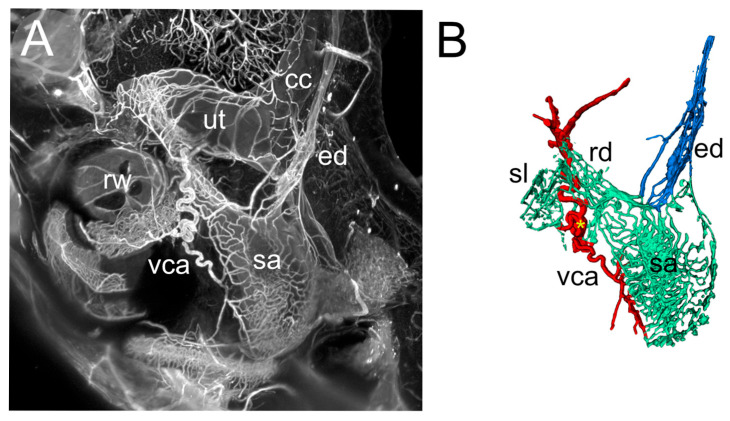
Vascular network associated with inner ear ducts. (**A**): Maximal intensity projection of a 200 µm stack from ColIV signal showing the saccule vascular network. (**B**): Volumetric reconstruction of the structures shown in (**A**) (the spiral limbus was removed in (**A**) for clarity). The yellow asterisk marks the branching of the perilymphatic straight vessel. The VCA is visible as a tortuous vessel associated with the RD, whereas the ED receives venous inflow. The two ducts are therefore different regarding their vascular surrounding. Abbreviations: cc: crus commune; ed: endolymphatic duct; rd: reuniting duct; rw: round window; sa: saccule; sl: spiral limbus; ut: utricle; vca: vestibulocochlear artery.

**Figure 5 audiolres-15-00160-f005:**
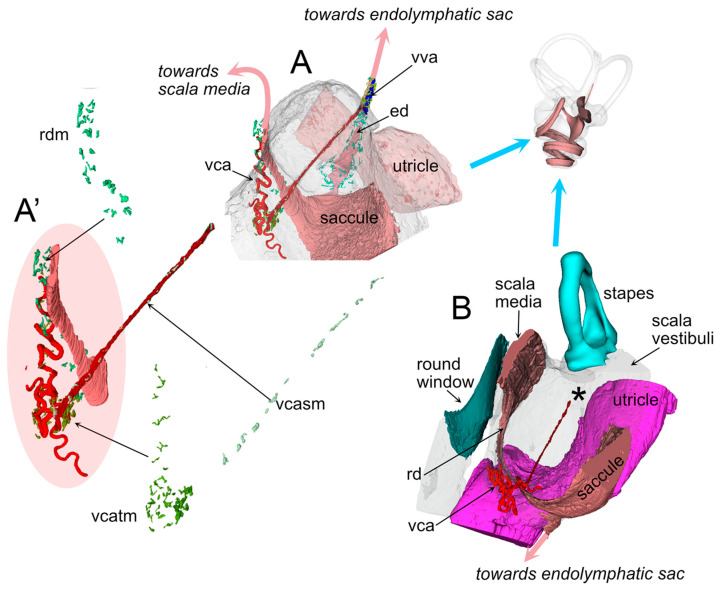
Volumetric reconstruction of macrophage populations associated with vessels and RD. Panel (**A**) displays the scala vestibuli cistern with saccule and utricle in their positions, and the macrophage populations associated with RD, VCA, and endolymphatic sinus. (**A’**) shows a magnification of RD, VCA, and their associated macrophages. Panel (**B**) displays the same region as A, but rotated in order to show the position of RD and VCA relative to the round and oval window. Asterisk marks the position of the utricular limiting membrane (not shown for image clarity; see [14] for depictions of the structure). The perilymphatic vessel is exposed to scala vestibuli pressure changes due to stapes movements. Abbreviations: ed: endolymphatic duct; rd: reuniting duct; rdm: RD macrophages; vca: vestibulocochlear artery; vcatm: vestibulocochlear artery (tortuous) macrophages; vcasm: vestibulocochlear artery (straight) macrophages; vva: vein of the vestibular aqueduct.

**Figure 6 audiolres-15-00160-f006:**
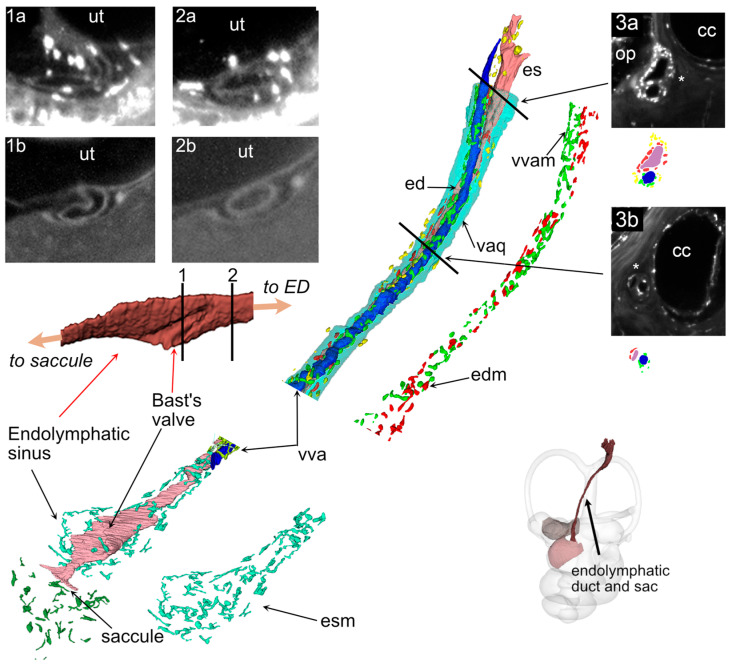
ED. Main panel: Volumetric reconstruction of the vestibular aqueduct (**top**) and endolymphatic sinus (**bottom**) with related macrophages. The endolymphatic sinus is displayed from the utricular face (in both reconstructions) to show Bast’s valve (visible as a fold in the sinus). The black lines on the top sinus reconstruction indicate the planes of optical sections indicated above. Macrophages associated with the sinus are shown in cyan, those associated with the ED and VVA in red and green, respectively. Macrophages not adjacent to either ED or VVA are shown in yellow. Panels 1 and 2 show optical sections at the level of Bast’s valve (1) and sinus (2), from the Iba1 signal (**1a**,**2a**) or from autofluorescence (**1b**,**2b**). Autofluorescence and Iba signals are from two different samples. Iba1+ cells (macrophages) are present in close association with the valve. Panel 3 shows optical sections at the level of the proximal sac (**3a**) and mid-duct (**3b**). Sac and duct are marked by asterisks. Below each section, the relative segmentation is shown, with the same color scheme as the main panel. Abbreviations: cc: crus commune; ed: endolymphatic duct; edm: ED-associated macrophages; es: endolymphatic sac; esm: endolymphatic sinus macrophages; op: operculum; ut: utricle; vva: vein of the vestibular aqueduct; vvam: VVA-associated macrophages; vaq: vestibular aqueduct.

**Table 1 audiolres-15-00160-t001:** List of temporal bone samples.

Sample	Age (Days)	Sex	Signal	Voxel Size (µm)
R1a,b	111	F	Auto, Iba1	2
R2	95	F	Auto, Iba1	2
R3	494	M	Auto, SMA	3.26
R4	79	M	ColIV	4.08

**Table 2 audiolres-15-00160-t002:** Macrophage population sizes in a single sample.

Solidity	Volume	Radius (um)	Number	Region
0.5 ± 0.2	5393 ± 3059	10.5 ± 2.0	301	All
0.4 ± 0.2	6066 ± 2885	11.0 ± 1.7	56	Endolymphatic sinus
0.7 ± 0.1	8266 ± 4385	12.2 ± 2.1	51	ED
0.5 ± 0.1	7267 ± 3785	11.7 ± 2.0	18	RD
0.5 ± 0.2	3748 ± 2162	9.3 ± 1.8	48	VCA
0.6 ± 0.2	8122 ± 4440	12.1 ± 2.2	63	VVA
0.7 ± 0.1	7404 ± 3218	11.9 ± 1.7	65	Vestibular aqueduct

## Data Availability

Data are available from the Authors upon request.

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
