# Peer review of "Reuniting and Endolymphatic Duct Macrophages: Localization and Possible Roles"

_audiolres, 2025, doi:10.3390/audiolres15060160_

Round 1
Reviewer 1 Report
Comments and Suggestions for Authors
I would like to begin by acknowledging what I found genuinely impressive about this work. The tissue clearing combined with light-sheet microscopy has yielded images of remarkable technical quality, and I must say the 3D reconstructions are visually compelling. The anatomical description of macrophage networks particularly their relationship to Bast's valverepresents what appears to be a novel observation that could prove relevant for our understanding of inner ear immunobiology. This aspect clearly demonstrates both technical expertise and considerable effort.
That said, I find myself compelled to raise serious concerns about the central scientific claims, which I believe are not adequately supported by the data presented.
Major Concerns
I was initially drawn to the promise of quantitative analysis highlighted prominently in the title, abstract, and results sections. However, as I examined the methodology more carefully particularly Table 2. I was quite surprised to discover that these "quantitative" analyses rest entirely on a single specimen (n=1). While I recognize the technical challenges inherent in this type of work, applying Gaussian mixture models or any statistical framework to a dataset of one strikes me as fundamentally problematic. Rather than creating genuine quantification, this approach risks giving an appearance of statistical rigor that may mislead readers. In its current form, I would characterize this as closer to an elaborate case report than a reproducible quantitative study.
What also concerns me is how the small sample (n=4) introduces multiple uncontrolled variables: different markers (Iba-1 vs. SMA/ColIV), different ages, and varying imaging conditions. Having reviewed similar studies over the years, I've seen how combining such heterogeneous specimens into a single dataset can undermine comparative interpretation. While the resulting figures are undeniably attractive, I question whether they can be taken as coherent biological evidence.
As I worked through the results and discussion, I became increasingly troubled by instances where single-animal observations are presented as if they represent general biological principles. The proposed differences between endolymphatic duct and vestibular macrophages, for example, appear to be built from insufficient comparisons yet are then linked to transcriptomic categories and even clinical outcomes. These connections strike me as speculative leaps that deserve to be clearly flagged as preliminary hypotheses rather than supported conclusions.
This manuscript illustrates what I see as a concerning mismatch between technical innovation and biological inference. While the imaging approach is genuinely excellent, I remain unconvinced that the dataset justifies the level of interpretation offered. My worry is that the work risks overstating its significance and potentially misleading readers about the robustness of its findings.
From my perspective, the real contribution here appears to be descriptive: an impressive demonstration of technical feasibility and an intriguing first glimpse at anatomical features that warrant further investigation. If the authors were to reframe their work explicitly in these terms without the overstatement of quantitative claims. I believe it could provide a valuable foundation for future research.
I cannot recommend acceptance in the current form, though I want to emphasize that this reflects concerns about framing rather than dismissal of the technical achievement. I would strongly encourage the authors to consider:
- Recasting the paper as a descriptive anatomical and imaging study
- Being more transparent about the inherent limitations of single-specimen data
- Presenting the quantitative methods (e.g., Gaussian models) as exploratory tools rather than definitive analytical frameworks
With these adjustments, I believe this work could still make a meaningful contribution by demonstrating technical possibilities and establishing a foundation for future studies with more robust sample sizes. The imaging approach alone merits publication it simply needs to be presented with appropriate scope and limitations clearly acknowledged.
Author Response
Agreed. We reframed the paper (including its title, which does not refer to quantification any longer) as a descriptive study and emphasized limitations due to sample size and technical issues. We performed nonparametric statistics only, which are not affected by data normality. We also simplified and clarified figures and methods, and added scripts as supplemental data. The language has also been checked and improved.
Reviewer 2 Report
Comments and Suggestions for Authors
This paper contains much new interesting information about the population of macrophages in the rat inner ear. Unfortunately the paper is not well written and needs major revision. This is unlike their former paper. The manuscript should be language edited thoroughly. RD is generally named reunion duct instead of reuniting duct.
Firstly in the abstract the M&M section does not give the reader a well explanation what is done and which technique has been employed. It should contain a more clear information that the authors used microCT investigation of cleared samples using lightsheet imaging after decalcification. Several abbreviations have to be explained clearly. There are several wrong statements saying initially that saccular otoconia has been linked to endolymphatic hydrops and is a key pathological hypothesis for Ménière’s disease. It is one key hypothesis.
Figure 1 is very good though the reunion duct is hard to see in A2. Figure 4 is very intriguing but hard to understand. Is it really possible to distinguish two populations of macrophages in the tiny vestibular aqueduct where the vein and duct are so closely situated? I think the image should be modified and the macrophages be colored unitedly. Explanation to abbreviations are not given in the legend. Figure 4B is confusing and the bright cells undefined. Bast valve looks unfamiliar.
Solidity of macrophages is interesting to better understand and measure cell spread and different macrophage phenotypes but should be better explained such as ratio of cell area to convex area. Solidity calculation as shown at line 107 are not possible to understand.
Relevant references are cited. Is SMA explained?
Conclusion though is well formulated and is intriguing! “The presence of macrophage populations associated with RD and ED and associated vessels suggests that they may affect endolymph circulation in the labyrinth. Macrophages associated with RD may sense local pressure variations and be involved in removal of otoconia debris plugging duct lumen. Macrophages associated with ED appear different and related to the venous rather than arterial compartment. However, their role, especially around Bast’s valve, although intriguing, remains obscure.”

Very interesting new data on the inner ear macrophages and how they could influence endolymph pressure and flow. Language should be improved. Recommend a professional language editing.
Author Response
This paper contains much new interesting information about the population of macrophages in the rat inner ear. Unfortunately the paper is not well written and needs major revision. This is unlike their former paper. The manuscript should be language edited thoroughly. RD is generally named reunion duct instead of reuniting duct.
We have revised language throughout the paper. As for the RD name, we followed naming from (Smith et al. 2022). We added both names when addressing the acronym.
Firstly in the abstract the M&M section does not give the reader a well explanation what is done and which technique has been employed. It should contain a more clear information that the authors used microCT investigation of cleared samples using lightsheet imaging after decalcification. Several abbreviations have to be explained clearly. There are several wrong statements saying initially that saccular otoconia has been linked to endolymphatic hydrops and is a key pathological hypothesis for Ménière’s disease. It is one key hypothesis.
Abstract has been revised. We did not use microCT in the present paper, just lightsheet scans. Abbreviations have been added to figure captions. We toned down the possible link between saccular otoconia, hydrops and Meniere, stating that it is one of the possible hypotheses.
Figure 1 is very good though the reunion duct is hard to see in A2. Figure 4 is very intriguing but hard to understand. Is it really possible to distinguish two populations of macrophages in the tiny vestibular aqueduct where the vein and duct are so closely situated? I think the image should be modified and the macrophages be colored unitedly. Explanation to abbreviations are not given in the legend. Figure 4B is confusing and the bright cells undefined. Bast valve looks unfamiliar.
We have added a section view of the duct segmentation in Figure 1B, and a high contrast inset for figure 1A2. Figure 1 is now Figure 2, since we moved solidity depiction to Methods as Figure 1.
In Figure 4A, (now 6A) the rings of macrophages around endolymphatic duct and vein have a partial overlap, and have now been lumped together (we kept different colors for image clarity). From the shown sections it is however observable that they outline both structures. Molecular and functional data will be needed to assess possible differences. We removed color overlay in the two displayed sections, to better show the actual images. Figure 4B (now 6B) has been expanded to show both Bast's valve and a section of the endolymphatic sinus, we hope it is clearer now. Orthogonal sections of Bast’s valve from the same sample used for panels 1b and 2b are shown in a previous paper (https://doi.org/10.7717/peerj.19512/fig-5)
Solidity of macrophages is interesting to better understand and measure cell spread and different macrophage phenotypes but should be better explained such as ratio of cell area to convex area. Solidity calculation as shown at line 107 are not possible to understand.
We clarified it in the methods, also adding a figure, and apologize for writing it backwards in the original submission. Calculations are performed with the enclosed MATLAB script
Relevant references are cited. Is SMA explained?
We added reference for SMA labeling of arteries.
Conclusion though is well formulated and is intriguing! “The presence of macrophage populations associated with RD and ED and associated vessels suggests that they may affect endolymph circulation in the labyrinth. Macrophages associated with RD may sense local pressure variations and be involved in removal of otoconia debris plugging duct lumen. Macrophages associated with ED appear different and related to the venous rather than arterial compartment. However, their role, especially around Bast’s valve, although intriguing, remains obscure.”
Comments on the Quality of English Language
Very interesting new data on the inner ear macrophages and how they could influence endolymph pressure and flow. Language should be improved. Recommend a professional language editing.
We have revised language throughout the paper.
Reviewer 3 Report
Comments and Suggestions for Authors
The authors' use of the iDISCO+ technique for cochlear clearing and visualization of its fine vascular and macrophage structures is impressive. Their results excellently demonstrate the distribution of macrophages in the inner ear, including the vestibular aqueduct. This represents a notable technical advancement and holds significant value for subsequent investigations into macrophage functions.
In the discussion, the authors propose that macrophages around the reuniting and endolymphatic ducts might correspond to the CD206+CD163+ macrophage population identified in Chiot et al.'s study on cochlear macrophage subpopulations, suggesting a potential role in regulating inner ear fluid homeostasis. Given that distinct macrophage subpopulations indeed exhibit different functions, and the authors observed differences in the solidity of macrophages in the reuniting duct (RD) and endolymphatic duct (ED), this further underscores the importance and urgency of verifying the specific subpopulations of these macrophages.
In this study, the authors named macrophages based on their different locations. Previous literature has reported the presence of perivascular macrophages (PVMs) in regions such as the stria vascularis of the inner ear. Could the reuniting and endolymphatic duct macrophages discussed by the authors also be a type of PVMs? It is recommended to add an analysis of the positional relationship between macrophages and blood vessels. If possible, a comparison with macrophages in the stria vascularis could be included.
The labeling of the reuniting duct (RD) structure in Figure 1 needs to be more detailed. If feasible, please add a cross-sectional image of the RD connecting the cochlear duct and the saccule to show the degree of patency of the duct.
Clinical research evidence regarding inner ear macrophages in patients with Ménière’s disease (MD) should be supplemented. Background information related to the "inner ear immune microenvironment in MD patients" (e.g., recent clinical studies reporting elevated inflammatory factors in the endolymph and positive macrophage markers in MD patients) should be added to provide more direct clinical evidence for the "link between macrophages, inflammation, and hydrops."
Image analysis methods: Although the authors mention that "three experts performed independent segmentations with a volume overlap > 90%," objective indicators such as the Kappa coefficient were not used to verify the consistency of segmentation.
Additionally, the MATLAB/R scripts used for calculating macrophage solidity were not made accessible (e.g., via a GitHub link), making it impossible to reproduce the analysis process.
It is recommended to add immunofluorescence co-labeling experiments and electron microscopy studies:
Perform co-labeling with Iba1 (macrophage marker), CD68 (activation marker), and otoconial proteins (e.g., OTOP1) to observe whether activated macrophages colocalize with otoconial debris, and use confocal microscopy to capture phagocytic phenomena.
Supplement transmission electron microscopy images to display the ultrastructure of macrophages phagocytosing otoconial debris (e.g., intracellular otoconial particles), thereby directly confirming the phagocytic function.
Author Response
In this study, the authors named macrophages based on their different locations. Previous literature has reported the presence of perivascular macrophages (PVMs) in regions such as the stria vascularis of the inner ear. Could the reuniting and endolymphatic duct macrophages discussed by the authors also be a type of PVMs? It is recommended to add an analysis of the positional relationship between macrophages and blood vessels. If possible, a comparison with macrophages in the stria vascularis could be included.
Thanks for the deep question. Perivascular macrophages are known to play a unique role in the brain, and vascular districts in the ear are so specialized that it appears very likely the associated macrophages are also specialized. However, for both RD and ED, we observed macrophage clusters directly associated to macroscopic vessels and other clusters associated to ducts. Unfortunately, we cannot answer to the question of their nature (similar or different) at the moment: further studies with additional markers will be needed. We were able to colocalize macrophages and macrovessels thanks to vessel autofluorescence or SMA coupled to Iba1 signal. Capillaries, on the other hand, did not show enough autofluorescence to be visualized without immunolabeling for collagen IV, and antibodies for collagen IV and Iba1 in our cleared samples could not be used for colocalization (being both made in rabbit). So, given the density of capillaries in the limbus and RD connective tissue, we cannot exclude that even the RD macrophages that are not associated to macrovessels are not perivascular. For ED, we now show clearer sections where it appears visible that macrophages surround the vein and duct in one single layer, but since the endolymphatic duct is a unique structure we do not know whether it could be associated to vascular-type macrophages. On the other hand, the stria vascularis often exhibited high autofluorescence and nonspecific labeling with Iba1, thus hindering the analysis of strial macrophages. We are working on these issues, which will be addressed in a second paper. As a response to comments from all reviewers, we have decided to tone down the present paper, leaving out most quantification (which will be addressed in further work with a larger sample population and additional markers) and focusing on the descriptive microanatomy.
The labeling of the reuniting duct (RD) structure in Figure 1 needs to be more detailed. If feasible, please add a cross-sectional image of the RD connecting the cochlear duct and the saccule to show the degree of patency of the duct.
We have added more detail now, including a section of the 3D reconstruction at the level of RD in figure 1B (now 2B). Additional details on RD geometry can be found in our previous paper https://doi.org/10.7717/peerj.19512 especially in Figure 3.
Clinical research evidence regarding inner ear macrophages in patients with Ménière’s disease (MD) should be supplemented. Background information related to the "inner ear immune microenvironment in MD patients" (e.g., recent clinical studies reporting elevated inflammatory factors in the endolymph and positive macrophage markers in MD patients) should be added to provide more direct clinical evidence for the "link between macrophages, inflammation, and hydrops."
references have been added
Image analysis methods: Although the authors mention that "three experts performed independent segmentations with a volume overlap > 90%," objective indicators such as the Kappa coefficient were not used to verify the consistency of segmentation.
Kappa coefficient has been added.
Additionally, the MATLAB/R scripts used for calculating macrophage solidity were not made accessible (e.g., via a GitHub link), making it impossible to reproduce the analysis process.
We added scripts as supplementary files.
It is recommended to add immunofluorescence co-labeling experiments and electron microscopy studies:
Perform co-labeling with Iba1 (macrophage marker), CD68 (activation marker), and otoconial proteins (e.g., OTOP1) to observe whether activated macrophages colocalize with otoconial debris, and use confocal microscopy to capture phagocytic phenomena.
Supplement transmission electron microscopy images to display the ultrastructure of macrophages phagocytosing otoconial debris (e.g., intracellular otoconial particles), thereby directly confirming the phagocytic function.
This will be done in further work.
Round 2
Reviewer 1 Report
Comments and Suggestions for Authors
Thank you for your message and for sharing the authors’ point-by-point response and the revised manuscript. I have carefully reviewed both. The revision satisfactorily reframes the study as descriptive and tempers causal claims. I recommend Minor Revisions prior to acceptance, limited to the following:
- Table 2. Please remove significance asterisks and p-values, as the cell-level observations derive from a single animal. Report descriptive statistics only (e.g., median/IQR). If the authors wish to retain tests, they should move them to the Supplement with an explicit note that they are exploratory, n=1 animal, no inferential claims, and remove “significant” language from the main text and captions accordingly.
- Abstract (final sentence). Please soften the functional wording so it is clearly hypothetical. For example:“Macrophages are observed in positions consistent with potential roles in sensing luminal changes and in the clearance of obstructive material; functional confirmation will require targeted experiments.”
(Ensure consistent wording in the Results/Discussion.) - With these adjustments, I will be pleased to recommend acceptance.
Author Response
Thank you for your comments. We have softened the abstract conclusions as you suggested and removed statistical considerations from Methods (paragraph about Kruskal-Wallis and Dunn test), Table 2 and main text (removing mentions to "significative" differences in Results and Discussion).
Reviewer 2 Report
Comments and Suggestions for Authors
The paper is much improved and ready for publication.
Author Response
Thank you